# You Can Train from Scratch:
# Further Discussion on the Long Range Arena

## Abstract

Despite their success, Transformers suffer from quadratic complexity in the sequence length, limiting their applicability to long-range dependency problems and making them expensive to train and run. After many proposals to address this issue, the Long Range Arena (LRA) was suggested as a benchmark to evaluate the performance of new models in long-range dependency modeling tasks. The Transformer and its variants performed poorly on this benchmark, and a new series of architectures such as State Space Models (SSMs) gained some traction, greatly outperforming Transformers in the LRA. Recent work has shown that with a denoising pretraining phase, Transformers can achieve competitive results in the LRA with these new architectures. In this work, we discuss and explain the superiority of architectures such as MEGA and SSMs in the Long Range Arena, as well as the recent improvement in the results of Transformers, pointing to the positional and local nature of the tasks. We show that while the LRA is a benchmark for long-range dependency modeling, in reality most of the performance comes from short-range dependencies. By using rotary embeddings and training techniques to mitigate its data inefficiency, the Transformer is also able to reach state-of-the-art performance without a separate pretraining phase. What is more, with the same techniques, we are able to remove all restrictions from SSM convolutional kernels and learn fully parameterized convolutions without decreasing performance, suggesting that the design choices behind SSMs merely added inductive biases and learning efficiency for these particular tasks. Our insights indicate that LRA results should be interpreted with caution and call for a redesign of the benchmark.

## 1 Introduction

The Transformer architecture (Vaswani et al.) has revolutionized sequence modeling, and has been the basis for many state-of-the-art breakthroughs in NLP, computer vision, and reinforcement learning. Despite its success, it is not without limitations. Being reliant on the attention mechanism, the Transformer suffers from quadratic complexity in the input sequence length, which makes it difficult to scale to long sequences and expensive to train and run. Addressing this issue has become a hot topic in the deep learning community, and many variations of the Transformer have been proposed to improve its scalability.

In this work, we focus our attention on the Long Range Arena benchmark (Tay et al.), which was designed to evaluate the ability of models to capture long-range dependencies. This benchmark encompasses tasks in the image, text, and mathematical domains, and includes sequences of up to 16,384 elements. The Transformer architecture and its variants have been shown to perform poorly on these tasks, which has motivated the development of new architectures.

These new architectures, such as State Space Models (SSMs) (Gu et al., b), were synthesized to a common underlying idea by Li et al.. They can all be reformulated as a long-convolution, with a kernel size matching the full sequence length. This kernel should be efficiently parameterized, that is, scale sublinearly with the input sequence length, and should include a time decay mechanism, reducing the weight of elements based on their distance to the current token.

Although these architectures greatly outperformed the Transformer on the Long Range Arena benchmark, the success of the Transformer in real-world settings has not been replicated. This raises

Table 1: Accuracy comparison between different kernel sizes for a convolutional model on the LRA tasks. For a given kernel size $K$, each embedding aggregates information from $\lfloor K/2 \rfloor$ embeddings to each side from the previous layer (second column). Thus, by multiplying by the number of layers (second to last row), we get the maximum range of dependencies that can be modeled. The accuracy of the model is shown together with the ratio of performance with respect to the results from the state-of-the-art model MEGA (last row). This ratio is shown in parentheses. In bold we highlight the best kernel size for each task.

| Kernel size | Perc. field | CIFAR10 | Pathfinder | Text classification | Text retrieval | ListOps |
|---|---|---|---|---|---|---|
| 5 | 2 | 46.63% (51.56%) | 51.80% (53.95%) | 88.96% (98.37%) | 90.32% (98.98%) | 45.24% (71.65%) |
| 7 | 3 | 49.05% (54.23%) | 51.42% (53.56%) | **90.61% (100.20%)** | **90.48% (99.16%)** | 49.26% (78.02%) |
| 11 | 5 | 64.47% (71.28%) | 51.56% (53.70%) | 90.56% (100.14%) | 90.41% (99.08%) | 50.59% (80.12%) |
| 21 | 10 | 70.25% (77.68%) | 52.01% (54.17%) | 90.26% (99.81%) | 89.71% (98.31%) | 52.28% (82.80%) |
| 31 | 15 | 79.84% (88.28%) | 51.20% (53.33%) | 89.33% (98.78%) | 89.46% (98.04%) | **52.93% (83.83%)** |
| 61 | 30 | **83.62% (92.46%)** | **72.28% (75.28%)** | 85.41% (94.45%) | 89.04% (97.58%) | 52.73% (83.51%) |
| Num layers: | | 6 | 6 | 4 | 4 | 6 |
| MEGA results: | | 90.44% | 96.01% | 90.43% | 91.25% | 63.14% |

questions about how representative the LRA benchmark results are of long-range dependency modeling performance. Indeed, the time-decay mechanism that these new architectures have in common suggests a bias towards locality, which is counterintuitive to modeling long-range dependencies.

Recent work by Amos et al. has shown that, in fact, the Transformer can achieve competitive performance on the LRA benchmark with appropriate training strategies. In particular, they used a pretraining phase with a denoising objective, using the same data as the downstream tasks. In this work, we achieve similar results without the need for a separate pretraining phase, reducing the computational burden and the risk of representation collapse during the fine-tuning stage. We use different data augmentation strategies for the image and mathematical domains, and a denoising objective in a multitask learning setting for the text domain. We show that with these strategies, the Transformer can achieve competitive performance on the LRA benchmark. We hypothesize the reasons for the necessity of these strategies in the case of the Transformer: very high-dimensional and insufficient data, and a very expressive architecture with poor inductive biases for the tasks.

Amos et al. showed that when using their pretraining strategy, a vanilla diagonal linear RNN can achieve state-of-the-art performance on the LRA benchmark. We further show that, using our strategies, an unrestricted and fully parameterized long convolution can achieve similar results as well. This suggests that restrictions to the kernel such as those synthesized by Li et al. are not necessary when enriching the data with proper strategies, but were only a way to add inductive biases and learn more efficiently in these particular tasks.

We discuss the reasons behind these results, arguing that the tasks are mainly positional and local. We measure the importance of locality and short-range dependencies by training convolutional models of increasing receptive fields. As shown in table 1, with small kernels of size $61$ we get close to state-of-the-art results in all tasks. Particularly concerning is the case of text, where kernels of size $5$ suffice. This heavily favors long-convolution-based architectures with time-decay mechanisms. Rotary embeddings add similar biases to the Transformer, and our ablation study shows its critical importance in achieving state-of-the-art results in the LRA.

Our results call the validity of the LRA as a long-range dependency modeling benchmark into question and indicate that it is important to analyze performance on the LRA with caution, taking into account the inductive biases of the models and the nature of the tasks.

Our contributions can be summarized as follows, in order of importance:

- We discuss the positional and local nature of the tasks in the LRA benchmark, and provide empirical evidence that one can reach close to state-of-the-art results with small convolutions that bound the range of dependencies that can be modeled. This explains the great performance of models such as MEGA or SSMs, and the poor performance of Transformers. It also questions the validity of the LRA as a long-range dependency modeling benchmark.
- We show that the restrictions on the kernel synthesized by Li et al. are not necessary when enriching the data or training process with proper strategies, but merely added inductive biases and learning efficiency in these particular tasks.

- We show that the reason for the poor performance of Transformers was a lack of inductive biases, coupled with its high expressiveness and high-dimensional, scarce data. With rotary embeddings that add positional and local biases, and techniques to avoid overfitting, we are able to reach state-of-the-art performance. Unlike the techniques used by Amos et al., ours do not require a separate pretraining phase, which reduces the computational burden and eliminates the risk of representation collapse.

The code is publicly available at `https://anonymous.4open.science/r/paper-LRA-source-anon-D370`.

## 2 BACKGROUND

### 2.1 THE ATTENTION MECHANISM

The attention mechanism, paramount in the Transformer architecture, suffers from quadratic complexity in the sequence length. Indeed, given queries $Q \in \mathbb{R}^{L_1 \times D_{qk}}$, keys $K \in \mathbb{R}^{L_2 \times D_{qk}}$, and values $V \in \mathbb{R}^{L_2 \times D_v}$, where $L_1$ and $L_2$ are the sequence lengths, and $D_{qk}$ and $D_v$ are the dimensions of the query/key and value vectors, respectively, the output of the attention module is given by

$$Y = \text{softmax}\left(\frac{QK^T}{\sqrt{D_{qk}}}\right)V. \tag{1}$$

The attention matrix $A = \text{softmax}\left(QK^T/\sqrt{D_{qk}}\right)$ is of size $L_1 \times L_2$, yielding a computation time complexity of $O(L_1 L_2 D_{qk})$ and a naive space complexity of $O(L_1 L_2)$. In reality, we can avoid quadratic memory complexity by recomputing the attention matrix during the backward pass (Dao et al.).

Reducing the cost of running the Transformer and allowing it to process long sequences is a hot topic in Deep Learning. From hardware usage optimizations such as the I/O optimizations in Flash Attention (Dao et al.) or precision reductions and quantization (Gholami et al.), to architectural modifications like the Linear Transformer (Katharopoulos et al.), the Reformer (Kitaev et al.), blockwise attention (Qiu et al.; Dai et al.) or sliding-window attention (Zaheer et al.), we have seen a large body of work on this very topic.

### 2.2 LONG RANGE ARENA

To evaluate the performance and efficiency of the different architectural proposals for long-sequence modeling, Tay et al. proposed a benchmark with tasks in the text, image, and math domains, called the Long Range Arena. These tasks were designed to include long sequences, ranging from 1000 to 16000 elements, while keeping the computational burden reasonably low. It includes six different datasets.

**ListOps** This synthetic task involves calculating the result of nested mathematical operations in parentheses, which form a tree-like structure. Both operands and results are always integers between 0 and 9, and sequences can be as long as 2000 tokens between operations and operands. Four different operations can appear: minimum (`MIN`), maximum (`MAX`), median (`MED`), and sum modulo 10 (`SM`). The following is a short example: $\text{MIN}\left[\text{MAX}\left[1, 2, 3\right], 4, 5\right] = 3$.

**Text Classification** The IMDB sentiment analysis task, involving the classification of movie reviews as positive or negative. In this benchmark, proposals are required to tokenize the text at the byte level, instead of the usual subword tokenization. They should also use sequence lengths of 1000, 2000, 3000 or 4000 bytes.

**Document Retrieval** This task involves finding a similarity score between two documents. The dataset is the ACL Anthology Network, where the model has to predict whether two papers have a citation link. The documents are tokenized at the byte level as well, this time requiring a fixed sequence length of 4000 bytes. Inter-token interaction between documents is not allowed for this task, this is, each document needs to be summarised to a fixed-size vector representation and then compared.

**Image Classification** The CIFAR-10 image classification task. Images are $32 \times 32$ pixels, and the task is to classify them into one of ten classes: airplane, automobile, bird, cat, deer, dog, frog, horse, ship, and truck. In this benchmark, images are given in grayscale, and they must be encoded as a one-dimensional sequence of individual pixels, totaling $32 \times 32 = 1024$ elements. No information about the two-dimensional structure of the image can be provided to the model.

**Pathfinder** A synthetic task that involves identifying whether two white points in a black background are connected by a path of dashed white lines. The images are $32 \times 32$, and the encoding restrictions are the same as in the Image Classification task.

**Pathfinder-X** An extreme version of the previous task, with larger images of $128 \times 128 = 16384$ pixels.

The Transformer achieved very poor results in this benchmark, both in the vanilla version and in the proposed variants to handle long sequences. In fact, results across variants were fairly similar in most tasks. The largest differences were found in the ListOps task, and we offer an explanation in Appendix B.

## 2.3 NEW ARCHITECTURES

Motivated by the poor performance of the Transformer in the Long Range Arena, several works have been published proposing new architectures that greatly improved their performance and computational efficiency. In this section, we review some of the most relevant.

### 2.3.1 STATE SPACE MODELS

In 2022, Gu et al. (b) proposed a new architecture, S4, based on the discretization of a differential equation model called the State Space Model (SSM). For a 1D input signal $u(t)$, the differential equation is the following:

$$\begin{cases} x'(t) = Ax(t) + Bu(t) \\ y(t) = Cx(t) + Du(t) \end{cases}, \tag{2}$$

where $x$ is the hidden state, $u$ is the input signal, and $y$ is the output signal. The elements $A$, $B$, $C$, and $D$ are the parameters of the model. The system is discretized with a fixed time step $\Delta$, and the $D$ parameter is removed, as it can be seen as a skip connection. The discretized system is then:

$$\begin{cases} x_{t+1} = \hat{A}x_t + \hat{B}u_t \\ y_t = Cx_t \end{cases}, \tag{3}$$

where $\hat{A} = f_A(\Delta, A, B)$ and $\hat{B} = f_B(\Delta, A, B)$ depend on the discretization method. This equation can be seen as a linear recurrent network without inter-token nonlinear activations. This greatly reduces expressiveness, as nonlinearities will only appear in token-wise operations. However, it also has great properties. In addition to the recurrent formulation that allows for very fast auto-regressive inference, the recurrence can be expanded to

$$x_{t+1} = \sum_{i=0}^{t} \hat{A}^i \hat{B} u_{t-i},$$

and thus be seen as a convolution operation, allowing for fast and parallelizable training.

When the input signal is a vector, the parameters $A$, $B$, and $C$ are matrices. In this case, the exponential of the matrix $A$ is not straightforward, but the kernel can be efficiently computed by parameterizing the matrix $A$ as the sum of a normal diagonal matrix minus a low-rank matrix. This architecture pushed the state-of-the-art average accuracy in the Long Range Arena from $59.37\%$ to $86.09\%$.

### 2.3.2 SYNTHESIZING SSMs

Posterior work has been published that synthesizes the key ideas that make the S4 model work. Orvieto et al. developed a similar linear recurrent model with a diagonal parameterization with

complex eigenvalues in the unit disk. They thus removed the discretization step or the normal minus low-rank decomposition. Furthermore, independent processing of each dimension in the diagonalized space allows us to process the convolution in $\mathcal{O}(L \log L)$ by using the Fast Fourier Transform, where $L$ is the sequence length. They achieved performance comparable in the long range arena to that of the S4 model.

Note that S4 has a convolutional mode that uses a kernel of the same size as the input signal. Li et al. tried to generalize these architectures by creating a convolutional model directly and trying to pin down the properties that these long convolutions should have to achieve good performance. They found two sufficient conditions. First, the kernel should be parameterized in a way that makes the effective number of parameters scale sublinearly with sequence length. Second, the kernel should satisfy a decaying structure, that is, the weights assigned to distant tokens should be smaller in magnitude than those assigned to closer tokens. They developed the SGConv architecture, which is a simple convolutional model that satisfies these properties, and achieved comparable performance in the Long Range Arena. The authors showed that the S4 model fulfilled these properties, and the model developed by Orvieto et al. clearly does too, as the exponential of a matrix with complex eigenvalues of norm less than 1 will decay in magnitude, and the number of parameters is linear in the hidden dimension and independent of the sequence length.

### 2.3.3 MEGA

At the time of writing, the current state of the art in the Long Range Arena is MEGA (Ma et al.), which alternates between the attention mechanism and an exponential moving average (EMA) across the sequence length. More formally, given input embeddings $x_1, \ldots, x_L$ in $\mathbb{R}^D$, the output of the EMA component is

$$y_t = \alpha \odot x_t + (1 - \alpha \odot \delta) \odot y_{t-1}, \tag{4}$$

where $\delta \in (0,1)^D$ is a damping vector and $\alpha \in (0,1)^D$ is a component-wise decay factor. The EMA operation is also a form of diagonal linear recurrence with eigenvalues of norm less than 1, like the one in Orvieto et al.. This yields a convolutional kernel that meets the properties discussed by Li et al.. The MEGA model achieved state-of-the-art performance in the Long Range Arena, with an average accuracy of $88.21\%$.

### 2.4 Improving the Transformer in the Long Range Arena

Amos et al. managed to achieve close to state-of-the-art performance in the Long Range Arena with the rotary Transformer by pretraining the models with a denoising objective, using the data from the downstream task. Similar to the pretraining of BERT (Devlin et al.), the model is asked to predict randomly masked tokens in the sequence.

Using this pretraining procedure, they also achieved competitive performance with a simple diagonal linear recurrent model, without the need for any complex parameterization or initialization as in the S4 model or the model by Orvieto et al..

### 2.5 gMLP

Liu et al. tried to achieve BERT's performance on standard text tasks using only MLP networks. To do this, they developed the *gMLP* architecture, alternating between channel-wise and spatial-wise or sequence-wise projections. Let $X \in \mathbb{R}^{L \times D}$ be the input data, where $L$ is the sequence length and $D$ is the embedding dimension. The gMLP layer is defined as follows:

$$Z = \sigma(XU), \quad Z_1, Z_2 = \text{split}(Z, \text{axis=``D''}), \quad \hat{Z} = Z_1 \odot (W Z_2 + b), \quad O = \hat{Z}V, \tag{5}$$

where $U \in \mathbb{R}^{D \times H}$ and $V \in \mathbb{R}^{H \times D}$ are projections in the channel dimension, acting on each token independently, and $W \in \mathbb{R}^{L \times L}$ and $b \in \mathbb{R}^L$ are the weights and biases of a linear projection across the sequence dimension. The function $\sigma$ is a non-linear activation function such as the GELU.

They managed to achieve competitive performance with BERT on the GLUE benchmark. The authors observed that the learned spatial projection matrices $W$ were, in fact, Toeplitz matrices, which resulted in a long convolutional operation. Unlike the long convolutions in the models described in the previous sections, the kernels in gMLP are fully learned and unrestricted.

## 3 TRAINING METHODS

As we hypothesized that the reasons for the poor performance of the Transformer in the LRA are the lack of inductive bias, and very high-dimensional and insufficient data, we apply techniques to avoid overfitting and prevent the model from learning spurious correlations.

**CIFAR-10.** Images are amenable to very natural data augmentation techniques. In the CIFAR-10 classification task, we apply a strategy derived from the one discovered in for AutoAugment (Cubuk et al.). The specific techniques can be found in the source code files indicated in appendix A.3.

**Pathfinder.** The LRA dataset provides three sets of examples of different difficulty, depending on the path lengths and amount of distracting paths. The common procedure is to train only on the difficult set. We try training on all three sets, without augmentation. The advantage of this over using augmentation is that the dataset is more varied and that the model is able to find the signal faster in the easier sets. Further discussion is provided in Appendix C.

To make the comparison fair and maintain the number of training steps, we reduce the number of epochs from the usual 200 to 67. Validation and test sets are drawn from the difficult set, and we remove subsets of similar sizes from the easy and intermediate ones to have exactly three times the number of training samples.

**ListOps.** As each mathematical operation in the task is commutative, we can shuffle the order of the operands without altering the result. In other words, at each node of the operation tree, we can apply a random permutation to the children nodes. Using this technique, we can generate many more training samples from the same input data.

**Text tasks.** Since data augmentation is not as straightforward in textual data, we apply a denoising objective where we mask some tokens and ask the model to predict them. This is similar to the masked language modeling objective in BERT (Devlin et al.) and what Amos et al. did in their work. However, instead of training for this task in a separate pretraining phase, we use a multi-task setting, where the total loss is the (unweighted) sum of the denoising loss and the task-specific loss. This has the advantage of reducing computational costs (as computations up to the final heads are shared), as well as removing the risk of representation collapse during fine-tuning.

Using this auxiliary task presents several advantages. First, since masking is done dynamically and randomly, the model sees different samples at each epoch, which helps to prevent overfitting. Second, a token-wise classification task such as the denoising one provides a large number of training samples per batch. Finally, since each text is tokenized at the byte level, the model needs to learn to aggregate neighboring bytes to form meaningful word embeddings. The MLM objective is known to produce a reliance on neighboring tokens to predict the masked ones, thus resulting in useful representations for the main task.

## 4 TRANSFORMER PERFORMANCE

To test whether the lackluster performance of Transformers stemmed from a lack of inductive biases and insufficient data, we train the Transformer using our mitigating strategies. Similarly to Amos et al., we use rotary embeddings (Su et al.) to encode positional information. We compare our results with the original Transformer results in the Long Range Arena (Tay et al.) and those obtained by Amos et al. using a denoising pretraining objective. We also include results from the state-of-the-art MEGA model (Ma et al.), and replicate their experiments using our training techniques.

The results are shown in table 2. We achieved a comparable performance with Amos et al. without the need for a pretraining phase, reaching a higher average accuracy and outperforming in all non-textual tasks. Both sets of techniques manage to take the Transformer to near state-of-the-art results. We also observe that our techniques do not seem to improve the performance of MEGA. In fact, it gets slightly worse, possibly due to an inexact replication of their optimization process to match ours (see appendix A.2).

Our results suggest that prior Transformer results were indeed due to a lack of inductive bias for the tasks and insufficient data, as we hypothesized. The MEGA model, on the other hand, was able

Table 2: Accuracy comparison between training a Transformer model on the LRA using a denoising pretraining objective, using our training techniques and the original results from Tay et al. without techniques. We also report the original results from MEGA (Ma et al.) and replicate their experiments with our techniques. Results for the PathX task are not available due to a lack of computational resources, and they are also excluded from the average calculation to allow for a fair comparison.

| Task | CIFAR10 | Pathfinder | Text classification | Text retrieval | ListOps | Average |
|---|---|---|---|---|---|---|
| Transformer (Tay et al.) | 42.44% | 71.40% | 64.27% | 57.46% | 36.37% | 54.39% |
| Transformer[†](Amos et al.) | 86.04% | 94.16% | **91.02%** | **91.57%** | 61.49% | 84.86% |
| Transformer[‡] | 88.15% | **96.28%** | 90.33% | 90.80% | 62.90% | 85.69% |
| MEGA (Ma et al.) | **90.44%** | 96.01% | 90.43% | 91.25% | **63.14%** | **86.25%** |
| MEGA[‡] | 87.60% | 94.78% | 90.92% | 91.32% | 61.15% | 85.15% |

[†] Using pretraining.     [‡] Using our training techniques.

Table 3: Accuracy comparison between gMLP and SSMs S4 and S5. We use our training techniques to train all three models, and also include the original results for S4 and S5. Results for the PathX task are not available due to a lack of computational resources, and they are also excluded from the average calculation to allow for a fair comparison.

| Task | CIFAR10 | Pathfinder | Text classification | Text retrieval | ListOps | Average |
|---|---|---|---|---|---|---|
| S4 (Gu et al., a) | 88.65% | 94.20% | 86.82% | 90.90% | 59.60% | 84.03% |
| S5 (Smith et al.) | 88.00% | 95.33% | 89.31% | **91.28%** | 62.15% | 85.21% |
| S4[†] | 89.59% | 93.61% | **91.01%** | 90.73% | 56.00% | 84.19% |
| S5[†] | 85.53% | 95.60% | 90.21% | 88.96% | **62.75%** | 84.61% |
| gMLP[†] | **89.89%** | **97.26%** | 89.94% | 90.37% | 62.45% | **85.98%** |

[†] Using our training techniques.

to reach its peak performance without better training strategies, as its inductive biases make it very data-efficient in this benchmark.

## 5 REMOVING RESTRICTIONS FROM LONG-CONVOLUTION KERNELS

Next, we use the same techniques to train an unrestricted and freely parameterized long-convolution-based model, gMLP, and compare it with SSMs S4 (Gu et al., a) and S5 (Smith et al.). The results are shown in table 3. The gMLP model achieves performance comparable to both SSM models and even outperforms them on average. The SSM models barely benefit from improved training strategies, suggesting that they were able to reach peak performance without them because of their inductive biases. Our results indicate that the design choices in SSMs merely added data efficiency for these particular tasks, and not better long-range dependency modeling.

## 6 DISCUSSION ON THE POSITIONAL AND LOCAL BIASES OF THE LONG RANGE ARENA TASKS

Our results indicate that the differences in performance between the Transformer and new architectures such as SSMs in the LRA are likely a byproduct of insufficient datasets and the greater inductive biases of the latter. If we look closely at each task, we can deduce that they are mainly positional. In particular, we expect a strong encoding of relative positions to be effective and more efficient to learn. Further, some of the tasks might benefit greatly from a bias towards locality. Both characteristics favor convolutional models with time decay mechanisms. This might also attribute some of the increased performance of the Transformer to the use of rotary embeddings, which adds similar biases.

Let us first consider textual data. Tokenizing text at the byte level means that the model has to learn first how to form meaningful word embeddings from letters, a positional and local task. Something similar can be said for images, where we often want to detect patterns in small patches (local

Table 4: Ablation study of positional encoding and training procedure for the Transformer on the LRA tasks. We show accuracy results for summed learned and sinusoidal positional embeddings, as well as rotary embeddings without using our training procedures.

| Positional embeddings Training techniques | Summed learned ✓ | Summed sinusoidal ✓ | Rotary ✗ | Rotary ✓ |
|---|---|---|---|---|
| CIFAR10 | 67.87% | 85.52% | 58.84% | 88.15% |
| Pathfinder | 90.48% | 92.81% | 67.09% | 96.28% |
| Text classification | 64.49% | 88.47% | 69.73% | 90.33% |
| Text retrieval | 83.31% | 87.06% | 85.34% | 90.80% |
| ListOps | 41.30% | 47.55% | 43.90% | 62.90% |
| Average | 69.49% | 80.28% | 64.98% | 85.69% |

neighborhoods). This is made slightly more difficult by the fact that we encode the image as a 1D sequence, separating consecutive vertical pixels by the width of the image, but the argument is still valid. This does not necessarily mean that long-range dependencies are not important. For example, in the Pathfinder task, the two connected dots can be very far apart in the image. Regardless of this, modeling relative positions is likely to be better and more efficient, as it is necessary to find neighboring pixels. Finally, in appendix B we provide a discussion of some of the patterns that are learned in the ListOps task. The extent to which local patterns account for the increase in accuracy to 63% escapes our analysis and still needs to be studied empirically.

## 7 ABLATION STUDY: TRANSFORMER PERFORMANCE

In this section we study the importance of correct positional encoding in increasing the performance of the Transformer. Table 4 shows the results. We compare rotary, summed sinusoidal and summed learned embeddings. The learned embeddings are very inferior to the other two, and rotary embeddings are superior to the summed sinusoidal ones, with small margins except for the ListOps task. These results were expected because rotary and sinusoidal embeddings do not include trainable parameters, model relative positions, and add a time-decay mechanism. The superiority of rotary embeddings might come from the fact that they are applied only to key and query vectors, without distorting intermediate representations or value vectors. Summed embeddings add this distortion, and they get entangled with the rest of the information across layers. In table 4 we also find the impact of removing our training techniques. While rotary embeddings—together with better hyperparameters—clearly improve the originally reported performance by the Transformer, not using our training techniques causes a massive drop in accuracy.

## 8 MEASURING THE IMPORTANCE OF LOCALITY IN THE LONG RANGE ARENA

The fact that a locality bias could be helpful in the LRA is concerning in a benchmark for long-range dependency modeling. To discern the importance of locality on each task, we train a convolutional model with small kernel sizes. This experiment should provide a baseline for the performance we can expect with a fixed bound on the range of dependencies that can be modeled. The details of the model can be found in appendix A.2, and the results are shown in table 1.

In general, we are able to get fairly close to state-of-the-art performance in each task with a small distance bound of 30 tokens per layer. The Pathfinder task appears to be the least reliant on very short-range dependencies. The extreme cases are the textual tasks, where a distance bound of 2 tokens per layer is sufficient to achieve state-of-the-art performance. It appears that aggregating bytes into meaningful word embeddings is enough to make predictions about most of the samples.

Our results show the great importance of short-range dependencies in these tasks, especially in the textual ones. This raises doubts about the relationship between the results in the LRA and the ability to model long-range sequences, especially in the case of models with local inductive biases, including recent architectures such as SSMs or MEGA, and Transformers with positional encodings

that add time-decay mechanisms. The evaluation of models in the LRA should include an analysis of the fairness of the results based on the biases of the model and the nature of the benchmark.

# 9 CONCLUSIONS

Our study provides critical insights into the nature of the tasks in the Long-Range Arena (LRA) benchmark, challenging its utility for long-range dependency modeling evaluation. Specifically, we identify that the tasks in the LRA benchmark largely benefit from positional and local dependencies. Our experiments demonstrate that small convolutional networks—limited in the range of dependencies they model—can closely match state-of-the-art results. This finding explains the success of models like MEGA and structured state-space models (SSMs) on the benchmark and raises questions about the adequacy of the LRA in truly testing long-range dependencies.

We also establish that the constraints on kernel design, as suggested by Li et al., are not strictly necessary to achieve strong performance. Instead, enriched data and optimized training strategies can yield similar benefits, with the proposed kernel constraints mainly providing inductive biases and learning efficiency in specific tasks. This opens new pathways for more adaptable model designs for long-range dependency modeling.

Our analysis highlights that Transformers' historically weak performance on LRA tasks stems primarily from a lack of inductive biases, rather than an inability to model dependencies. By introducing rotary embeddings for local and positional biases and refining training to prevent overfitting, they also manage to achieve state-of-the-art results.

Our findings point to a significant re-evaluation of both model architectures and the LRA benchmark for long-range dependency modeling. We underscore the need for benchmarks that genuinely assess long-range dependencies, but leave the design of such a benchmark for future work. As general guidelines, synthetic tasks such as Pathfinder seem to be the most robust to short-range dependencies and allow us to modulate the range of dependencies required to solve the tasks. Natural language seems to be an inconvenient modality, as it may prove difficult to create tasks that require long-range dependency modeling but low computational resources, which was one of the intentions behind the LRA. The restriction to simplified synthetic languages is probably the best choice.

ACKNOWLEDGMENTS

Anonymized during review.

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

## A  EXPERIMENT DETAILS

### A.1  HARDWARE AND SOFTWARE

All experiments were run on a single NVIDIA GeForce RTX 3090 GPU with 24GB of memory or an NVIDIA GeForce RTX 3080 Ti GPU with 12GB of memory. We used PyTorch 2.1.0 with CUDA 12.2. The code for the experiments is available at `https://anonymous.4open.science/r/paper-LRA-source-anon-D370`. A Conda environment file can be found in the repository to reproduce the Python environment.

### A.2  HYPERPARAMETERS

In tables 5 to 7 we show the hyperparameters used to train the Transformer, the gMLP model, and the convolutional model on the LRA. We used the AdamW optimizer, with a learning rate scheduler that reduces the learning rate on training plateau and during the last 10% of the training epochs. The parameters of the Transformer apply regardless of the positional encoding, except those specific to rotary embeddings. The base frequency for sinusoidal embeddings is the same as for rotary embeddings.

With regard to the MEGA and SSM experiments, we generally replicated their hyperparameters, with the following exceptions.

- When the models did not converge, we reduced the learning rate.

- The attention mechanism in MEGA is restricted to the classical softmax one.

- We use the same optimizers, schedulers and training epochs that we used for the Transformer, gMLP and convolutional models.

- For S5, we used the full-glu activation in Pathfinder and text retrieval, as we obtained better results. We also cut down the epochs in text retrieval due to lack of time during the rebuttal period.

With respect to the convolutional model, it consists of several equal layers with residual connections. Each layer starts with a convolution that maps the input embeddings to $H$ channels, with kernel sizes varying between experiments to modify the bound of the distance between tokens that can interact with one another. After a non-linearity, a linear layer returns back to $D$ channels. The code for each layer can be found in the file `src/models/layers/res_conv.py`.

### A.3  DATA AUGMENTATION

The augmentation techniques can be consulted in the source code.

- The augmentation process for the CIFAR10 dataset is covered in the `src/utils/augmentation/cifar10_augmentation.py` and `src/data_loaders/cifar10.py` files.

- The augmentation process for the ListOps dataset is covered in `src/data_loaders/listops.py`.

Table 5: Hyperparameters used to train the Transformer on the LRA. The rotary disable half parameter determines whether half of the base frequencies are set to 0, disabling the rotation of those channels and making them independent of positional information. In the Pathfinder task, the number of epochs is 67 when using all three sets, and 200 otherwise.

|  | ListOps | CIFAR10 | Pathfinder | Text classification | Retrieval |
|---|---|---|---|---|---|
| Embed dim | 256 | 160 | 128 | 256 | 128 |
| Depth | 6 | 10 | 6 | 8 | 6 |
| Num attn heads | 4 | 4 | 4 | 4 | 4 |
| FF size | 512 | 320 | 256 | 512 | 256 |
| Dropout | 0 | 0 | 0 | 0 | 0 |
| Activation | ReLU | ReLU | ReLU | ReLU | ReLU |
| Norm | Layer | Layer | Layer | Layer | Layer |
| Prenorm | Yes | Yes | Yes | Yes | Yes |
| Rotary base freq | 10000 | 10000 | 10000 | 10000 | 10000 |
| Rotary disable half | No | Yes | No | Yes | Yes |
| Pooling | Mean | Mean | Mean | Mean | Mean |
| Learning rate | 0.0001 | 0.001 | 0.001 | 0.001 | 0.001 |
| Batch size | 64 | 48 | 128 | 16 | 64 |
| Weight decay | 0.05 | 0.05 | 0.05 | 0.01 | 0.01 |
| Max epochs | 120 | 200 | 67/200 | 80 | 60 |

Table 6: Hyperparameters used to train the gMLP model on the LRA. $H$ is the expanded number of dimensions where the spatial convolution operates. The number of independent channels is the number of channels of the spatial convolution kernel, which are repeated to reach $H$ channels.

|  | CIFAR10 | Pathfinder | Text classification | Text retrieval | ListOps |
|---|---|---|---|---|---|
| Depth | 10 | 6 | 6 | 6 | 6 |
| D (embed dim) | 128 | 128 | 128 | 128 | 128 |
| H (hidden dim) | 256 | 256 | 256 | 256 | 256 |
| Independent channels | 2 | 2 | 2 | 2 | 4 |
| Max sequence length | 1024 | 1024 | 4096 | 4096 | 2000 |
| Dropout | 0 | 0 | 0.1 | 0 | 0 |
| Norm | Layer | Layer | Layer | Layer | Layer |
| Activation | GELU | GELU | GELU | GELU | GELU |
| Pooling | Mean | Mean | Mean | Mean | Mean |
| Learning rate | 0.001 | 0.001 | 0.001 | 0.0003 | 0.001 |
| Batch size | 48 | 128 | 16 | 64 | 32 |
| Weight decay | 0.1 | 0.1 | 0.2 | 0.1 | 0.1 |
| Max epochs | 200 | 67 | 80 | 60 | 120 |

### A.4 MULTI-TASK LEARNING

In the multitask learning environment, we summed both the downstream task loss and the denoising loss with the same weight of 1. For the denoising task, we mask 30% of the tokens in the input sequence, of which a third are replaced by random tokens, and the rest are replaced by a special mask token.

## B SOME REMARKS ON THE LISTOPS TASK

Transformers achieved very poor results in the ListOps task, with different variants ranging between 17% and 36% accuracy. If we look at the distribution of labels depending on the root operation in fig. 1, we can see that the labels are not uniformly distributed except for the SM operation. When the root operation is a MIN, the distribution is strongly biased towards 0. The same happens with the MAX operation, but towards 9, and with the MED operation, but towards 4. If we always predict 0 or

Table 7: Hyperparameters used to train the convolutional model on the LRA. At each layer, a convolution is applied mapping embeddings of dimension $D$ to $H$ channels. After a non-linearity, a linear layer returns back to $D$ channels. The size of the kernel varies for each experiment.

|  | CIFAR10 | Pathfinder | Text classification | Text retrieval | ListOps |
|---|---|---|---|---|---|
| Depth | 6 | 6 | 4 | 4 | 6 |
| D (embed dim) | 128 | 128 | 128 | 128 | 128 |
| H (hidden dim) | 256 | 256 | 256 | 256 | 256 |
| Groups | 1 | 32 | 1 | 1 | 1 |
| Dropout | 0 | 0 | 0 | 0 | 0 |
| Norm | Layer | Layer | Layer | Layer | Layer |
| Activation | ReLU | ReLU | ReLU | ReLU | ReLU |
| Pooling | Mean | Mean | Mean | Mean | Mean |
| Learning rate | 0.001 | 0.001 | 0.001 | 0.0003 | 0.001 |
| Batch size | 64 | 128 | 16 | 64 | 32 |
| Weight decay | 0.1 | 0.1 | 0.2 | 0.1 | 0.1 |
| Max epochs | 200 | 67 | 80 | 60 | 120 |

9, we already get around 17% accuracy. If we predict 0, 4 or 9, based only on the root operation, we get around 36% accuracy. These baselines yield precisely the results that the different Transformer models achieved.

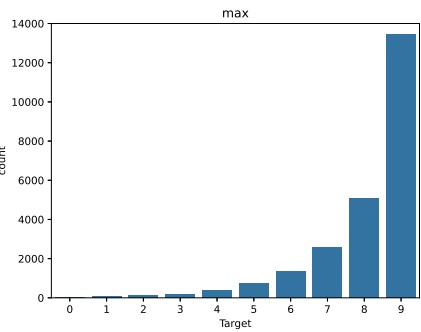

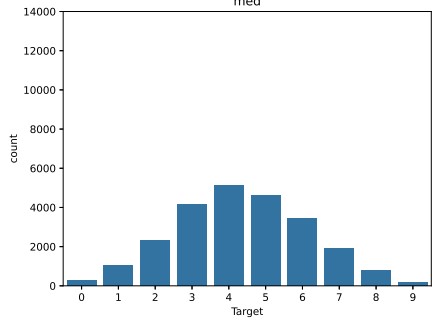

(a) Distribution of labels for examples with root MAX operation.

(b) Distribution of labels for examples with root MED operation.

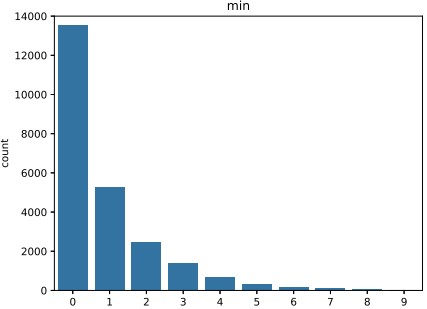

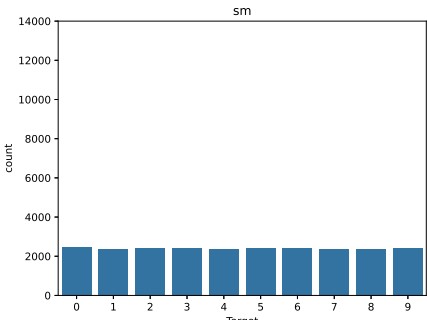

(c) Distribution of labels for examples with root MIN operation.

(d) Distribution of labels for examples with root SM operation.

Figure 1: Distribution of labels based on the root operations in the ListOps task.

We can also look at where the models that achieved the best results (around 63% accuracy) are improving their performance. In fig. 2, we compare the accuracy of a Transformer model that

achieves around 63% accuracy with a greedy algorithm that always predicts the most frequent label for each root operation (and randomly in the `SUM MOD` operation). We can see that the model is still guessing randomly in the `SUM MOD` operation, but it is greatly improving the accuracy in the rest of the operations.

In fig. 3, we can see the breakdown of accuracy by operation and label for the same Transformer model. It seems that the model is learning to discard the most frequent label for the `MIN`, `MAX` and `MED` operations, and moving its prediction upward or downward.

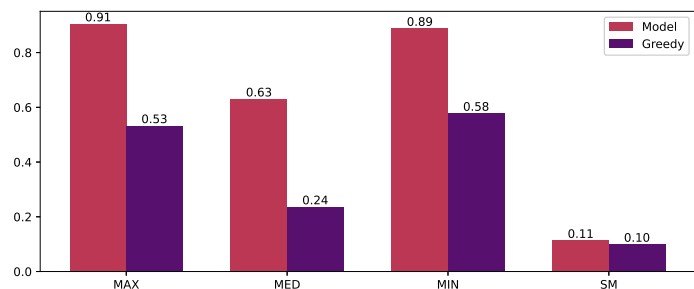

Figure 2: Accuracy difference between a Transformer model that achieves around 63% accuracy and a greedy algorithm that always predicts the most frequent label for each root operation (and randomly in the `SUM MOD` operation). The accuracies are reported by root operation.

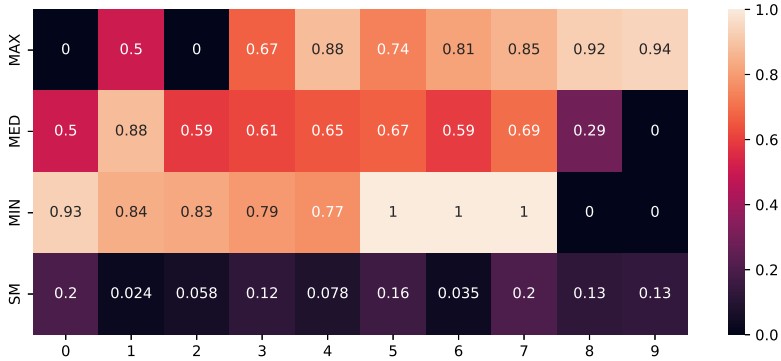

Figure 3: Accuracy breakdown by operation and label for a Transformer model that achieves around 63% accuracy in the ListOps task.

## C  REMARKS ON TRAINING FOR THE PATHFINDER TASK

For this task, we decided to learn from the easy and intermediate sets instead of using augmentation, or both. Of course, this is because we found better empirical results with this approach. The main consideration is that the Transformer seemed to need several passes over the most difficult examples to correctly learn them. The model seems to learn faster this way. However, not using any augmentation or other sets leads to overfitting. A good balance is having extra data or a *finite* set of possible augmentations. We recommend using a subset of the dihedral group $D_4$ of reflections and rotations, as they do not cause any distortion in the images due to low resolution. Depending on the ability of the model to overfit, one might decide to use more or less augmentations, as well as other training sets.

