# OpenReview forum: "You Can Train from Scratch: Further Discussion on the Long Range Arena"
_ICLR.cc/2025/Conference — Submitted to ICLR 2025_

### Official Review · Reviewer_3xaX · 2024-10-26

**Soundness:** 3
**Presentation:** 3
**Contribution:** 2
**Rating:** 5
**Confidence:** 4

**Summary:**

The authors analyze the Long-Range Arena dataset regarding the locality of relevant features for the prediciton tasks. They find that using very localized features in a convolution based architecture can give performances very close to SOTA, rendering Long-Range Arena effectively a "non-long-range" benchmark. They show how augmentation techniques can help achieving SOTA-like performance with Transformers using rotational positional embeddings - a technique well-known from other modalities (vision) for compensating the lack of inductive bias in Transformer models.

**Strengths:**

They can show that Long Range Arena can be "solved" to SOTA levels using very short convolution windows, rendering it a non-long range benchmark effectively.

**Weaknesses:**

The authors may analyze the Long-Range Arena dataset further in how it is actually not a long-range dataset.
The augmentation techniques used for reaching good performance on LRA with Transformers are well-known and are expected to benefit training.
For a clear accept / strong accept, I would expect a construction / assembly of a benchmark that actually tests long-range reasoning (beyond simple retrieval as in MQAR / AR / Needle in the Haystack tasks)

**Questions:**

Do you have ideas on how to improve upon the LRA dataset as a benchmark for long-range modeling?
Are there ideas for long-range tasks where (simple) augmentation techniques might fail?
The presentation of well-known Transformer and SSM architectures is unnecessary in that detail in my opinion.

---

> ### Author Response · Authors · 2024-11-14
>
> Thank you for your review! The criticism is fair, as it would have been a better contribution to provide a new benchmark that doesn't suffer from the problems we discussed. In the current version of the manuscript we focused on explaining the results of different models in the benchmark, and end with our main criticism of it being mostly reliant on short-range dependencies.
>
> Regarding the first part of your question, we would focus on creating synthetic tasks such as the Pathfinder one, which seems to be the most robust. We may be able to make it even more difficult by ensuring the lines coming out of a point always reach very far away. As for natural language, creating tasks that require long range dependency modeling and can be solved with models of moderate size (which was one of the points in creating the LRA) seems difficult. Again, the best option is to use tasks with synthetic and simplified languages. This is an interesting question, so we will try adding some indications about this in the conclusions!
>
> As for the final part of the question, our techniques mostly aid with overfitting, in a setting where the model is expressive enough to perform the tasks, but doesn't have enough data to find correct patterns instead of spurious correlations. These techniques should mostly help and not harm. They may fail to improve performance if the model is not expressive enough to fit the data, or if the model is not able to find the correct signal (except learning from easier examples, which might help in this case).

---

> > ### Comment · Reviewer_3xaX · 2024-11-25
> >
> > Thank you for your comments and insight. I hope to see your ideas explored in future work.

---

### Official Review · Reviewer_u525 · 2024-10-29

**Soundness:** 2
**Presentation:** 2
**Contribution:** 1
**Rating:** 1
**Confidence:** 5

**Summary:**

This paper explores the performance of Transformer models on the Long Range Arena (LRA) benchmark, which evaluates models' ability to handle long-range dependencies across diverse tasks in text, image, and mathematical domains. Traditionally, Transformers face challenges with quadratic complexity relative to sequence length, limiting their scalability and efficiency. While newer architectures like State Space Models (SSMs) have outperformed Transformers on the LRA, recent work by Amos et al. demonstrated that Transformers could achieve competitive results through a denoising pretraining phase. Building on this, the authors of this paper present training techniques that enable Transformers to attain similar or superior performance on the LRA without requiring a separate pretraining stage. These techniques include task-specific data augmentation, integrating a denoising objective within a multi-task learning framework for text tasks, and employing rotary embeddings for positional encoding. Through ablation studies, the paper reveals that many LRA tasks are predominantly positional and heavily reliant on short-range dependencies. This insight suggests that inductive biases favoring locality significantly enhance model performance. Additionally, the paper evaluates the gMLP architecture, demonstrating that unrestricted long-convolution-based models can surpass specialized architectures like SSMs on most LRA tasks. The authors conclude by cautioning that LRA benchmark results should be interpreted with an understanding of the models' inductive biases and the specific nature of the tasks, as these factors heavily influence performance outcomes.

**Strengths:**

The paper comprehensively analyzes the factors contributing to the Transformer's performance on the Long Range Arena benchmark. Exploring training techniques that eliminate the need for a separate pretraining phase addresses computational efficiency and simplifies the training pipeline.

**Weaknesses:**

* Novelty and originality: The paper does not introduce a new model. It leverages existing techniques, such as Rotary embeddings and data augmentation, making it purely applied / engineering work. The authors should clarify what they consider their work's main novelty or contribution beyond applying these existing methods.
* Depth of Analysis: The discussion provides valuable insights into the role of inductive biases and the characteristics of LRA tasks. However, it would benefit from a deeper exploration of these findings. Specifically, the authors should provide more detailed explanations of why certain training techniques are effective and theoretically explain how rotary embeddings improve performance. This additional analysis would strengthen the paper's contribution by offering a deeper understanding of the underlying factors influencing model performance.
* Poor experimental results: The evaluation is limited to the LRA dataset, which does not comprehensively demonstrate the models' capabilities on longer sequences. To improve the robustness of the findings, the authors should consider including additional benchmarks such as RULER (https://github.com/hsiehjackson/RULER) and the Path-X dataset from LRA. Additionally, updating their comparisons to include the most recent state-of-the-art results, not only MEGA, would provide a more current and thorough evaluation of their methods. Addressing these points would significantly improve the comprehensiveness and relevance of their experimental results.
* Already studied a lot: LRA has been studied a lot at this point and is not so important for long sequence modeling. SSMs solved it.

**Questions:**

No further questions.

---

> ### Author Response · Authors · 2024-11-14
>
> Thank you for your review! I am afraid the main points of our paper might not have been made as clear as possible, and I apologize for the inconvenience. The paper is mainly:
> - An explanation of the results different models have gotten in the LRA, explaining the positional and local biases that are favored in the tasks.
> - A critique of the LRA benchmark, as we show that short-range dependencies account for sufficient performance to reach near SOTA levels with small convolutional models.
> - A set of techniques that can be used to address the lack of biases in the Transformers (or in any other proposals) to make comparisons fairer.
>
> In light of this, I will respond to each of the listed weaknesses in order:
> 1. Indeed, the main contribution is not a new technique, but a critique of the benchmark and a questioning of the conclusions of previous work based on their results.
> 2. This is a great point! We tried to do it at the end of page 7: "These results were expected, as rotary and sinusoidal embeddings do not include trainable parameters, model relative positions, and add a time decay mechanism. The superiority of rotary embeddings might come from the fact that they are applied only to key and query vectors, without distorting intermediate representations or value vector". This was explained in the context of our critique: the importance of positional and local biases in the LRA. Perhaps we could have explained it better or more thoroughly.
> 3. We limit ourselves to the LRA precisely because we are trying to critique it. We are not showing the performance of our techniques or of our model, but explaining their results and previous results in this very benchmark.
> 4. Agreed! That's why we thought it would be a good idea to revisit it and its results, as the research line of SSMs initially originated around this benchmark. We were not trying to solve the LRA again, just explain the results of the different models and critique it.

---

> > ### Comment · Reviewer_u525 · 2024-11-25
> > **Reviewer u525 - keeping the same score**
> >
> > Thanks for the answers.
> >
> > Unfortunately, I will still keep the same rating as I do not think, in its current form, the paper is at a level close to the ICLR borderline.

---

### Official Review · Reviewer_ojL4 · 2024-10-29

**Soundness:** 2
**Presentation:** 3
**Contribution:** 2
**Rating:** 5
**Confidence:** 4

**Summary:**

The paper revisits prior results on the LRA benchmark and shows that rather than a separate pretraining phase, alternative training techniques can be applied to each task to achieve similar results with Transformers. Furthermore, the same training approach allows fully parameterized convolutions to match the performance of parameterized convolutions, such as SSMs. The authors provide a discussion on the importance of positional and locality bias in LRA tasks, showing that strong performance can be achieved with either a correct choice of positional embeddings or strong locality bias, despite the long range nature of the tasks.

**Strengths:**

* Benchmarks such as LRA are in widespread use for developing new architectures, questioning what are the key elements required to reach competitive results on them is an important question.

* The suggested approaches, of applying a different training strategy to each task, leads to competitive results - I especially like the approach in pathfinder, adding easier samples to induce a stronger signal early in training.

* The discussion in section 4.3 is insightful:

   * The importance of the correct choice of positional information is clearly explained.

   * The results for a convolution model in Table 2 are surprising and insightful, indicating importance of locality in the tasks.

**Weaknesses:**

* The broader implications of the main result, sections 4.1, 4.2 are unclear. If the claim is that LRA tasks are unrepresentative for performance on real world tasks it is unclear how results in 4.1,4.2 imply that. Alternatively, if the authors claim that their training setup is more realistic then prior work, it should be clearly stated and argued for with empirical results. If there are any additional implications I could not understand them from the text.

* Benchmarks such as LRA are designed to evaluate architecture inductive bias, introducing additional biases via other training strategies, such as augmentations, couples the modeling and augmentation induced biases.

   * For CIFAR10 the augmentations are not clearly described, looking in cited work [1] it seems that the augmentations are:  “...A sub-policy consists of two operations, each operation being an image processing function such as translation, rotation, or shearing, and the probabilities and magnitudes with which the functions are applied.” These augmentations are based on the 2D structure of the input and provide additional signal for the model to recognize it - which is part of the purpose of this task [2].

   * SImilarly for ListOps, it is desirable from a sequence model to learn the permutation invariant nature of the task from the data.

* It is not clear if the same training approach applied to Transformers and gMLP is used on the baselines (MEGA, S4 etc.). Results seem to be the same as those in cited work but this is not clearly stated - whether the performance gap still persists when models are trained in the same manner is important to the claim that Transformers can match performance.

**Questions:**

* In the introduction, the important question about: “how representative the LRA benchmark results are of long-range dependency modeling performance” (4th paragraph) is raised but I could not understand if it is explored later in the text - if it is not addressed, results connecting or disconnecting performance on LRA and real-world tasks can benefit the paper.

* Throughout the text it is mentioned that avoiding the pretraining phase reduces the risk of representation collapse - why does representation collapse matters when training on a single downstream task and reaching competitive results?

* In section 2.5 equation 5 the what is the definition of the `split` operation

* In table 2, Can you clarify the calculation of the receptive field? How does a kernel with size 61 have a receptive field of 30 tokens?

[1]  Ekin D. Cubuk, et. al.  Autoaugment: Learning augmentation strategies from data

[2] Yi Tay et. al LONG RANGE ARENA: A BENCHMARK FOR EFFICIENT TRANSFORMERS

---

> ### Author Response · Authors · 2024-11-14
>
> Thank you for your review! I will first answer to the listed weaknesses:
> - Your are correct! It is true that the implications are not clearly stated, and we will improve the clarity of the text for the final version. The summary of the implications is the following. The bad results from Transformers and the good results of models such as SSMs are due to their inductive biases, and they can be mitigated with techniques such as ours. The part about Transformers was already achieved in the paper by Amos et al., but they didn't discuss the reasons or do ablation studies on the importance of using rotary embeddings.
> - I agree that it would be ideal to not have to use special training techniques. However, the main point of the benchmark is to evaluate long range dependency modeling, and not data efficiency. Specially in this case were data efficiency is achieved by a locality bias in a benchmark that is supposed to evaluate long range dependency modeling.
> - This is a fair point! We will try to get those results ready for the final version of the manuscript.
>
> Now, regarding the questions:
> 1. We believe the experiments with small convolutions (table 2) answer that question. They show that most of the tasks can be "solved" to near SOTA levels with short-range dependencies, which means that models are not actually required to learn long-range dependencies.
> 2. You are absolutely correct! If empirically the results are stable and fine-tuning doesn't cause trouble, then it is not very relevant. We only stated it because our methods cannot suffer from that, whereas pretraining theoretically could. Other advantages such as computational resources and ease of programming are still advantageous.
> 3. Just splitting across the embedding dimension in two chunks of the same size (like in the GLU activation function).
> 4. Absolutely. It means that in the (k+1)-th layer, each embedding only has information about embeddings in the k-th layer with a relative distance of 30 or less. This is because the kernel of size 61 touches 30 tokens to the left, the center one, and 30 tokens to the right. I will try to make this clearer in the document. Thanks for the comment!

---

> > ### Comment · Reviewer_ojL4 · 2024-11-25
> >
> > The authors have acknowledged some of the stated weaknesses but I did not see any changes to the manuscript, hence I am keeping my score.

---

> > > ### Comment · Reviewer_ojL4 · 2024-11-26
> > >
> > > I went over the uploaded revision (which was published after my last comment).
> > > I think the updates, specifically results on SSMs showing that data augmentations don't maintain a considerable advantage over transformers, are meaningful and I update my score.
> > >
> > > Yet, I still think that the broader implications are lacking. If the main claim is about the local nature of the tasks, this is only addressed in a single section (the importance of PEs has been noted in prior work as other reviews mentioned).
> > > Furthermore, I still don't understand why using said training techniques is more representative of the performance on real-world tasks than results in original LRA.

---

> > > > ### Author Response · Authors · 2024-11-26
> > > >
> > > > Thank you for taking the time to read the revision! I understand your opinion, and I will try to give our point of view.
> > > >
> > > > Regarding the importance of PEs, are you referring to the paper by Amos et al.? I believe they did not address the importance of using Rotary Embeddings (could be mistaken), although they did use them. We did ablate their importance and tried to give an explanation about why they are so advantageous in these particular tasks.
> > > >
> > > > And regarding your final question, given our experiments we believe that the LRA results are not very representative either way when the model adds a locality bias, especially the textual tasks. Still, what we ideally want to measure is the ability to model long-range dependencies, and not data efficiency (although it would be desirable if it did not depend on locality). If we could find a sub-quadratic model that is able to model the data (without locality biases) but is not data efficient, it would still be an interesting option to pursue. Thus, we believe that standardizing techniques to decouple data efficiency from the results is important.

---

### Official Review · Reviewer_kqn3 · 2024-11-04

**Soundness:** 2
**Presentation:** 3
**Contribution:** 2
**Rating:** 5
**Confidence:** 4

**Summary:**

This work focuses on the performance of Transformer architecture on the Long Range Arena benchmark (LRA). Earlier works have shown that simple state-space / recurrent architectures can achieve better performance than Transformer architectures on LRA benchmark, while transformers are still the preferred choice in real-world sequential dependency modelling. Although Transformers can achieve on-par performance on LRA using a pre-training phase with a denoising objective, this leads to additional computational cost and risk of representation collapse. This works shows that Transformers can achieve similar performance on LRA benchmark without an additional pretraining stage with the help of better data augmentation strategies and rotary positional embeddings. This work argues that the tasks in LRA have a positional bias and LRA benchmark should be interpreted with caution.

**Strengths:**

- Incorporating data augmentation during transformer training helps avoid the pre-training stage and still achieves comparable results as state-of-the-art models for LRA benchmark.
- Ablations on positional embeddings show that transformers improve in performance with better positional embeddings.
- This work points out flaws in the popular LRA benchmark and urges to use this benchmark with caution.

**Weaknesses:**

- LRA has been known to have local positional bias in the literature (see R1, R2). R2 already incorporates some form of positional embeddings in the transformer architecture helps achieve better performance on LRA benchmark.
- Data augmentation would help improve performance of any model on this benchmark
- Proposed techniques in this work are marginal since pre-training in R2 already shows the objective which is used in this paper along with positional embeddings. Similarly, data augmentation techniques are well known in the literature to improve model performance.

--------
- [R1] ON THE LONG RANGE ABILITIES OF TRANSFORMERS : https://arxiv.org/pdf/2311.16620
- [R2] Never Train from Scratch: FAIR COMPARISON OF LONGSEQUENCE MODELS REQUIRES DATA-DRIVEN PRIORS : https://arxiv.org/pdf/2310.02980

**Questions:**

- This work mentions the risk of representation collapse in the two stage training, do you have any concrete evidence that such a collapse happens in your experiments?
- Do you know why gMLP does not improve on text-retrieval task with your setup in Sec. 4.2?
- Does the same conclusion hold true for other long range dependency benchmarks such as SCROLLS?
- Can you provide an estimate of resources required for training on Pathfinder vs Path-X tasks?
- Can you compare the performance of MEGA, S4, S5 when these models are trained with data augmentation strategies mentioned in this work?
- In Figure 2, why is only gMLP trained with the proposed data augmentation strategies? Why weren't these techniques applied on S4/S5?

---

> ### Author Response · Authors · 2024-11-14
>
> Thank you for your review! The addressed weaknesses are very fair.
> - Indeed, this was already suggested in prior literature. We cite R2 specifically, as we achieve similar results for the Transformer with other techniques. The aim was to give a thorough explanation of these results and the good performance of SSMs, and to analyze to what extent locality is important and sufficient to solve the tasks.
> - We used the techniques for the models that required it to avoid overfitting. Based on preliminary experimentation we saw that for the MEGA model it barely improved results. We will try to address this in the final manuscript by adding the results you asked for in the last two questions.
> - We agree that the techniques we used to train the Transformer and GMLP are not a major improvement over the techniques in R2, although they do have the advantages we listed: less computational resources and no risk of representation collapse. It is also simpler to code and set up. It should also be stated that the objective is only similar in textual tasks, and techniques for the rest of tasks are different.
>
> I now respond to each of the listed questions:
> 1. We did not seek empirical evidence of this. What we wanted to state is that our methods rule out that possibility while pretraining does not. It would indeed add strength to show empirically the possible instability of pretraining, but I am not sure we will have time to add that.
> 2. We are not sure. We didn't assign much significance to it, and thus didn't study it further. Any minor difference in the training process could reasonably explain this, but this is speculative and we do not have any evidence.
> 3. We didn't study it, but at first glance of the tasks I would say the conclusions do not hold for SCROLLS.
> 4. It multiplies the sequence length by 16,  and thus the total time by 256 in quadratic model such as the Transformer or MEGA. With our GPU and the Transformer we get over 12 hours for each epoch, so over one month for a full 67 epoch run.
> 5. The last two questions are fair points! We will try to get these results ready in the final version of the manuscript.

---

> > ### Comment · Reviewer_kqn3 · 2024-12-02
> >
> > Thank you for your reply. Given the current state of the work, I am keeping my score as the work needs improvements since many of the strategies (positional embeddings, data-augmentation) proposed has already been used to improve the performance of transformer models on LRA. It would also help to impact of the work by adding more datasets which evaluate long term dependence learning.

---

> > > ### Author Response · Authors · 2024-12-03
> > >
> > > Thank you again for your response. I understand that evaluating contributions can be subjective. However, I am afraid that there might be a misunderstanding about the actual contribution of the paper.
> > >
> > > To the best of our knowledge, other authors do not use data augmentation. But that is not very important, as our main contribution with that section is ablating and explaining the prior poor results from Transformers and their recent improvement.
> > >
> > > Regarding the final request, the point was not to evaluate a model on long-range dependency modeling. Rather, we are evaluating the validity of the LRA as a long-range dependency modeling benchmark. Thus, evaluating the models on other datasets would not contribute to the points we are trying to make.
> > >
> > > In any case, thank you for your input and good luck with your own submissions if you had any!

---

### Author Response · Authors · 2024-11-25

Hi everyone! Sorry for reuploading late into the revision period, but new experiments took some time. I wanted to thank you for your kind and insightful reviews, as I think they have greatly improved the paper. I will list now an overview of the main changes we made:

- We changed the abstract, conclusions and final part of the introduction to better explain and emphasize the contributions of the paper.
- To this end, we also changed the section structure, creating a separate section for each set of experiments.
- We tried to state the implications of each set of experiments more clearly in their section.
- We added new results by training MEGA, S4 and S5 with our own training techniques. As we expected, they barely improved, as thanks to their inductive biases they already reached their peak performance without them. This addresses the fair points raised about whether using our techniques could improve the other architectures as well and grow the performance gap again.
- Moved the convolutional experiment results to the introduction to emphasize its importance.

We also made some minor changes from your requests:
* Clarified receptive field calculations.
* Added clarity about where to find the CIFAR augmentation we used: the appendix linked to files in the code we shared.

Finally, we corrected a minor mistake from the results in the first draft. In the table with the ablation study of positional encodings in the Transformer, we fixed results of text classification for rotary with our techniques (last col), which was wrong, from 90.80% to 90.33%, and the average calculation as well.

If you wish to read it again, you should mainly review the abstract, introduction, results sections (now sections 4-8) and conclusions.

Again, thank you for your time and effort,
The authors

---

### Meta-Review · Area_Chair_PTuz · 2024-12-20

**Metareview:**

The submission suffers from several critical weaknesses. First, it lacks novelty, as it primarily employs established techniques which are well-documented and have been extensively applied in the literature. The paper's scope is also narrow, focusing solely on the Long Range Arena (LRA), a benchmark already known to favor local biases over genuine long-range dependencies. Furthermore, the analysis of the results is weak. The paper fails to convincingly argue for the relevance of the proposed methods to real-world long-range dependency tasks, leaving the broader implications unclear. This is compounded by inconsistent benchmarking, as several key baselines are not re-evaluated with the proposed augmentations, undermining the validity of performance comparisons.

**Additional Comments On Reviewer Discussion:**

Reviewers remain unconvinced after the rebuttal and discussion phase.

---

### Decision · Program_Chairs · 2025-01-22

Reject